# Impact of Glucocorticoid Cumulative Doses in a Real-Life Cohort of Patients Affected by Giant Cell Arteritis

**DOI:** 10.3390/jcm11041034

**Published:** 2022-02-16

**Authors:** Paul Castan, Anael Dumont, Samuel Deshayes, Jonathan Boutemy, Nicolas Martin Silva, Gwénola Maigné, Alexandre Nguyen, Sophie Gallou, Audrey Sultan, Achille Aouba, Hubert de Boysson

**Affiliations:** 1Department of Internal Medicine, Caen University Hospital, Avenue de la Côte de Nacre, 14000 Caen, France; castan-p@chu-caen.fr (P.C.); dumont-an@chu-caen.fr (A.D.); deshayes-s@chu-caen.fr (S.D.); boutemy-j@chu-caen.fr (J.B.); martinsilva-n@chu-caen.fr (N.M.S.); maigne-g@chu-caen.fr (G.M.); nguyen-a@chu-caen.fr (A.N.); gallou-s@chu-caen.fr (S.G.); sultan-a@chu-caen.fr (A.S.); aouba-a@chu-caen.fr (A.A.); 2UFR de Santé, Caen University-Normandie, 14000 Caen, France

**Keywords:** giant cell arteritis, glucocorticoids, tolerance, cumulative doses, adverse events

## Abstract

Objectives: To describe the impact of cumulative glucocorticoid (GC) doses on related adverse events (AEs) in giant cell arteritis (GCA) in a real-life setting. Methods: The medical charts of the last 139 consecutive GCA patients followed in a tertiary centre were retrospectively analysed. The cumulative GC doses were calculated, and the main GC-related AEs were collected during the follow-up. Results: After a median follow-up duration of 35.6 (2–111) months, the median cumulative GC dose in the 139 patients was 9184 (1770–24,640) mg, and 131 patients (94%) presented at least one GC-related AE. Infections (63%) were the most frequently reported GC-related AE, followed by metabolic events (63%), including weight gain in 51% of them. Cardiovascular and neuropsychiatric events occurred in 51% and 47% of patients, respectively. Osteoporotic fractures, muscular involvement, digestive events, geriatric deterioration, skin fragility, ophthalmologic complications and hypokalaemia were reported in <35% of patients. Cardiovascular events (*p* = 0.01), osteoporotic fractures (*p* = 0.004), cataract occurrence (*p* = 0.03), weight gain (*p* = 0.04) and infections (*p* = 0.01) were significantly associated with GC cumulative doses > 9 g. Longer GC durations were associated with cataract occurrence (*p* = 0.01), weight gain (*p* = 0.03) and all-grade infections (*p* = 0.048), especially herpes zoster occurrence (*p* = 0.003). Neuropsychiatric and metabolic events appeared within the first months after GC introduction, whereas herpes zoster recurred, and most cardiovascular AEs emerged after 1 year. Geriatric events, especially osteoporotic fractures, occurred 2 years after GC introduction. Conclusion: This study highlights how frequent GC-related AEs are and the impact of prolonged GC and cumulative doses.

## 1. Introduction

Giant cell arteritis (GCA) is a large vessel vasculitis affecting the aorta and its collaterals, mostly vessels with cephalic destiny [1]. This is the most frequent form of vasculitis among patients over 50 years old. The clinical presentation mainly includes cephalic symptoms such as headaches, jaw claudication, scalp tenderness and/or ophthalmologic involvement that can lead to blindness. In addition to cranial symptoms, polymyalgia rheumatica (PMR) and large-vessel symptoms such as back pain or peripheral vascular claudication are not rare. The risk of acute ischaemic complications, especially ophthalmologic involvement with a risk of bilateralization, increases the severity of this vasculitis and thus pushes for the introduction of an effective initial treatment [2].

Glucocorticosteroids (GCs) are the primary treatment intention and allow for the rapid control of vessel inflammation and ischaemic consequences. The initial high doses are maintained until clinical and biological control is obtained. Thereafter, the GC dose is progressively reduced over weeks until treatment is finally stopped, often after 12–24 months [2,3]. Relapses are frequent during follow-up and might affect half of the patients [4]. A few studies also suggest that some disease patterns tend to develop a disease course with more relapses and more GC dependence [5,6]. The notion of GC dependence has no clearly established threshold or validated definition. However, the inability to decrease GC doses under a certain dose, usually between 7 and 15 milligrams per day of prednisone equivalent, without disease relapses is generally considered a GC-dependent disease. The adjunction of a GC-sparing immunosuppressant is recommended in this setting.

Despite their intrinsic efficiency in treating disease, GCs lead to multiple adverse events (AEs) and affect metabolism (diabetes mellitus, dyslipidaemia, weight gain with faciotruncular repartition), the cardiovascular system (arterial hypertension, arrhythmia, coronaropathy and cardiac failure), the neuropsychiatric spectrum (including insomnia, irritability or mood disorders), the osteomuscular system (osteopenia or osteoporosis, myopathy and tendinopathy), visual function (cataract and chronic glaucoma), immunity with susceptibility to infections, the digestive tractus (haemorrhage, ulcer, gastric pain) or skin [7,8].

Proven et al. [9] demonstrated in 2003 among a cohort of patients with GCA diagnosed between 1950 and 1991 that cumulative GC doses were more closely correlated with AE occurrence than high initial doses. Since the study by Proven et al. [9], many other studies have shown the relation between cumulative GC doses and GC-related AEs [10,11,12,13,14,15,16].

Despite this known toxicity over time, recent studies have shown that GC duration remains prolonged between 2 and 3 years for most patients and that cumulative GC doses have increased over the last 6 decades [17,18].

To date, few real-life studies have captured GC-related AEs in detail. We therefore conducted a retrospective observational monocentric study in a well-defined cohort of consecutive GCA patients. We aimed to assess the cumulative GC doses in each patient and analyse the occurrence of GC-related AEs during follow-up. We also analysed the potential links between cumulative GC doses, duration of GC intake and the time of detection of each GC-related AE.

## 2. Material and Methods

### 2.1. Patient Selection

This retrospective observational monocentric study was conducted among the 149 last consecutive patients diagnosed with GCA in the Department of Internal Medicine at the University Medical Center of Caen between 2013 and 2018. The inclusion period was stopped in 2018 to have enough follow-up time after GC introduction to capture potential late AEs. All data regarding patient characteristics were collected in a centralized database in our centre, and follow-up information was updated after each medical control in our department.

We included and analysed in this study all patients with (1) a diagnosis of GCA depending on the presence of a vasculitis demonstration on temporal artery biopsy or vascular imaging or a clinical presentation suggestive of GCA with a good response to GC and the absence of an alternative diagnosis during the follow-up in a few patients and (2) enough information in the medical files about the daily GC doses to calculate the cumulative doses.

We excluded 10 patients from the cohort with incomplete data during follow-up to calculate the cumulative GC doses. We thus analysed data from 139 patients. This study was conducted in compliance with good clinical practices and the Declaration of Helsinki principles. At the time of this study, our local ethics committee (Caen-CLERS) confirmed that, in accordance with French public health law (Art. L 1121-1-1, Art. L 1121-1-2), a formal approval from an ethics committee was not required for this type of observational study.

### 2.2. Studied Parameters

All data regarding demographics, clinical presentation and paraclinical work-up were retrieved from the centralized database.

The cumulative doses for each patient were calculated as follows:

All GC intakes were taken into consideration, reported as prednisone equivalent, even if the patient was previously treated for PMR before GCA;

From the medical prescriptions that were given to the patient at each follow-up medical control, we estimated the correspondence daily dose/number of days;

If a discordance occurred between the theoretical dose the patient should be at, at the moment of the consultation, and the actual dose reported in the consultation document, the actual dose prevailed. The estimation from the prescription was then corrected by changing the dates to ensure regular intervals without changing the corresponding doses;

If a major discordance occurred between two doses (e.g., misunderstanding from the patients between 15 and 35 mg/day), the smallest dose was considered;

If no information was available on progressive tapering (e.g., patient lost to follow-up at a daily dose of 5 milligrams), we considered the dose for the duration as it was written on the prescription and then assumed direct weaning without decreasing;

If a dose was maintained for a long time without more information later, the calculation was stopped at the last follow-up consultation without considering this long ongoing therapy;

We stopped the cumulative dose calculation at 30 June 2021 for patients whose follow-up was still ongoing.

We collected from the medical files all AEs possibly related to GC as long as the patient was receiving ongoing GC treatment. The AEs considered were cardiovascular (arrhythmia, cardiac failure, arterial hypertension, coronaropathy, stroke); digestive (haemorrhage, ulcer, gastric pain); neuropsychiatric (psychiatric event, insomnia, cognitive impairment); muscular atrophy; new osteoporotic fracture; geriatric (fall, loss of autonomy); cutaneous (skin fragility with purpura or bruise, delayed cicatrisation); ophthalmologic (cataract, chronic glaucoma); metabolic (diabetes mellitus, dyslipidaemia, weight gain); hypokalaemia; and infections. We also included the occurrence of any neoplasia assuming the possible role of long-term immunosuppression. For patients with a medical history of hypertension or diabetes mellitus, a GC-related AE was considered if a loss of equilibrium occurred upon GC therapy. Weight gain was defined as an increase of a minimum of 2 kg. For each adverse event, we recorded the time between GC introduction and detection. The Common Toxicity Criteria for Adverse Events (CTCAE) was used to grade the adverse events (scaling from 1 to 5), with the 5th grade corresponding to death [19].

However, we retrieved some AEs after GC discontinuation due to their persistent effect, especially regarding the cardiovascular system, assuming endothelial injuries from arterial hypertension still remained; osteoporotic fractures; geriatric events; cataracts assuming the lack of clearance in this tissue; and possible neoplasia. Skin complications were analysed until a year after GC discontinuation assuming that this delay was necessary for the skin to no longer be affected.

The cohort was then divided into two groups according to the cumulative GC doses, and the prevalence of GC-related AEs was compared. Since the median GC cumulative dose in the cohort was 9 g, we artificially chose this dose as the threshold to distinguish the two groups. We also performed a second analysis with three groups of cumulative doses, <8 g, between 8 and 12 g, and >12 g. To ensure that cumulative doses were proportional to GC durations, we performed the same comparisons between patients who received GC for less than 20 months, from 20 to 40 months and more than 40 months. The thresholds at 20 and 40 months were artificially chosen for this study. We finally tried to identify from the group comparisons those AEs whose occurrence seems to be related to increasing cumulative GC doses or increasing duration treatment.

### 2.3. Statistical Analysis

We used Fisher’s exact test, the Chi square test or the Mann–Whitney test according to the variable of interest when comparing two groups, and the Chi square test, homogeneity Chi square test or Kruskal–Wallis quantitative test for analysis of 3 groups. The statistical analyses were computed using JMP 9.0.1 (SAS Institute Inc., Cary, NC, USA). A *p* ≤ 0.05 defined statistical significance.

## 3. Results

### 3.1. Patients Characteristics at Baseline

The patients’ characteristics are detailed in Table 1. Among the 139 patients, 102 (73%) were women, and the median age at diagnosis was 72 (52–92) years old. Cranial symptoms were present in 112 patients (81%), ophthalmologic involvement in 40 patients (29%) and polymyalgia rheumatica in 51 patients (37%) at diagnosis. Vasculitis was demonstrated on temporal artery biopsy in 86 patients (62%) and on imaging in 40/126 patients (32%) who underwent large-vessel imaging.

### 3.2. Cumulative Doses of Glucocorticoids and Main Related Adverse Events

The details of GC management are indicated in Table 1. After a median follow-up duration of 35.6 (2–111) months, 97 (70%) patients were able to definitively discontinue GC (median duration: 22 (8–100) months). At the last follow-up, the cumulative GC dose in the 97 patients who discontinued GC was 8231 (2836–24,383) mg. Considering the 139 patients, at the last follow-up, the cumulative dose was 9184 (1770–24,640) mg.

Within the cohort, 131 patients (94%) presented at least one GC-related AE. The details of each GC-related AE are shown in Table 2. Infections (*n* = 88 patients; 63%) were the most frequently reported GC-related AE, with a first event occurring at a median delay post GC introduction of 11 (0.1–77) months. Metabolic events affected 87 (63%) patients, including weight gain in 71 (51%) of them. Cardiovascular and neuropsychiatric events occurred in 71 (51%) and 66 (47%) patients, respectively. Finally, osteoporotic fractures, muscular involvement, digestive events, geriatric deterioration, skin fragility, ophthalmologic complications and hypokalaemia were reported in <35% of patients.

### 3.3. Impact of Cumulative GC Doses on Adverse Event Occurrence

In Table 2, we compared patients according to whether they received less or more than 9 g of GC cumulative doses. The following AEs were significantly associated with GC cumulative doses > 9 g: any cardiovascular events (61% vs. 40% in patients receiving < 9 g, *p* = 0.01), especially arterial hypertension (45% vs. 26%, *p* = 0.02); osteoporotic fractures (22% vs. 5%, *p* = 0.004); cataract occurrence (26% vs. 11%, *p* = 0.03); weight gain (59% vs. 42%, *p* = 0.04); and all infections (73% vs. 52%, *p* = 0.01).

In Table 3, GC-related AEs were compared in three groups of GC cumulative doses (<8 g, between 8 and 12 g, and >12 g). Osteoporotic fractures (*p* = 0.009) increased with increasing GC cumulative doses.

### 3.4. Impact of GC Duration on Adverse Event Occurrence

At the last follow-up, 47 patients were receiving or had received GC for less than 20 months, 50 for 20 to 40 months, and 42 for more than 40 months. GC-related AEs according to GC duration are listed in Table 4. Longer GC durations were associated with cataract occurrence (*p* = 0.01), weight gain (*p* = 0.03) and infections (*p* = 0.048), especially herpes zoster occurrence (*p* = 0.003).

### 3.5. Impact of Immunosuppressant Use on Adverse Event Occurrence

In the cohort, 37 (27%) patients received an immunosuppressant, methotrexate or tocilizumab, in combination with GC. In Table 5, we compared the main AEs in patients who received or did not receive an immunosuppressant. Only arrhythmia was more frequently observed in patients who received an immunosuppressant (*p* = 0.004). There was no difference in either group regarding the other AEs, especially the occurrence of infections. In addition, we did not observe a lower rate of GC-related AEs in patients who received an immunosuppressant.

### 3.6. Delay between GC Introduction and GC-Related AE Occurrence

The median time of the first detection of each main AE is indicated in Figure 1. Neuropsychiatric and metabolic events such as diabetes mellitus or hypokalaemia were more frequently detected within the first months after GC introduction. The first grade 3–5 infections requiring hospitalization occurred earlier than other less severe infections at 6 and 10–11 months. Herpes zoster recurrences emerged one year post-GC introduction. Most cardiovascular AEs were detected after 12 months. Finally, geriatric events, especially loss of autonomy, falls and osteoporotic fractures, occurred 2 years after GC introduction.

## 4. Discussion

Our study provides real-life data about GC management in GCA and their related AEs. Since we deeply analysed patients’ medical files during the follow-up, we provided longitudinal unedited data about the impact of prolonged GC therapy. We observed frequent GC-related AEs in more than 90% of patients and showed the impact of increasing GC durations (and subsequent GC cumulative doses) on cardiovascular, osteoporotic and infectious complications. Some AEs seemed to be associated with either high cumulative doses or longer GC exposure. Metabolic and neuropsychiatric effects occurred rapidly after GC introduction, probably subsequently to the high daily doses. Infections and cardiovascular AEs were mainly observed after one year of treatment. Finally, geriatric complications with autonomy loss, falls and fractures appeared more frequently two years after GC introduction.

According to the EULAR recommendations dating from 2018, as well as the French GEFA guidelines from 2016, a daily dose of 15–20 milligrams/day should be reached at 2 to 3 months of GC therapy and then tapered to reach 5 milligrams at 12 months [3,20]. In the present study, these thresholds were not respected since the 20 mg and 5 mg thresholds were achieved after a median of 5 and 20 months, respectively. These differences might be explained by the frequent relapses and subsequent increases in GC doses during the follow-up. Moreover, during relapse, recommendations often advise increasing GC to the previous daily dose that controls the disease, but in real life, physicians often prescribe higher doses to obtain rapid control of the flare. Altogether, this probably explains why GC tapering was longer than recommended in the main guidelines.

Chandran et al. [17] showed some changes in the GC management of GCA patients over recent decades, with a trend to increase GC durations and related cumulative doses. Thus, the cumulative GC dose at 5 years of follow-up increased from 7.6 g in the period 1950–1979 to 10.7 g in 1980–2009, and the total GC duration increased from 1.5 to 2.6 years in these time periods. Other studies reported similar results on the increasing tendency of cumulative GC doses. Proven et al. [9], in a cohort dating from 1950 to 1994, reported a GC cumulative dose of 7.35 g, while a more recent study reported cumulative GC doses between 8 and 11 g [16,18]. Of note, some other studies have reported lower cumulative GC doses, but their register-based protocols may lack longitudinal data to collect all GC therapy information [10,15].

Many register-based studies have shown the association between GCA status and AE incidence [12,16] or directly between GC therapy and GC-related AEs [10,13,15]. The main AEs described in those register-based studies are retrieved via hospitalization medical codes that include cardiovascular events, diabetes mellitus, osteoporosis and fractures as well as serious infections requiring hospitalization. However, some other adverse events, such as myopathy, cataracts, glaucoma, hypokalaemia or herpes zoster, are rarely described. Albrecht et al. [14] focused on the prevalence of these previous AEs and found results similar to those in our study, with 1/5 of patients experiencing osteoporosis or cardiovascular events and 1/6 of patients with diabetes mellitus.

Contradictory data exist regarding the risks of GC-related AEs with GC doses < 5 milligrams/day. Strehl et al. [21] considered a daily dose of less than 5 milligrams per day to be relatively safe in terms of AE occurrence unless a high cardiovascular risk was detected. However, Pujades-Rodriguez et al. [11] demonstrated a persisting risk of cardiovascular AEs in patients with a daily dose of less than 5 milligrams. They also showed that 15% of patients suffering from inflammatory diseases with ongoing GC therapy had cardiovascular AEs. This result is inferior to our prevalence, probably due to their exclusion of patients having a medical history of cardiovascular diseases, a population who may experience decompensation with ongoing GC therapy.

Schmidt et al. [22] showed that severe infections occur preferentially during the first year of GC therapy, which is concordant with our median delay of occurrence at 6 months. Wu et al. [23] also reported similar results.

Our study design allowed us to provide some information regarding other GC-related AEs that are less frequently described, such as neuropsychiatric, geriatric and digestive events. However, neuropsychiatric or geriatric events were not associated with either increasing cumulative doses or longer GC duration. One explanation for neuropsychiatric events may be that those effects tend to appear with high initial daily doses and thus early in the disease course, therefore being less affected by long exposure. Geriatric events such as loss of autonomy and falls might lead to osteoporotic fractures, which are significantly associated with high cumulative GC doses, but detection of these geriatric events is difficult, as they often occur at patients’ homes before hospitalization and are not always mentioned in the medical files.

Our patients were treated in accordance with French and European guidelines, recommending the use of GC for at least 18 months [3,20]. With a high rate of GC-related AEs in our patients, this study underlines the urgent necessity of GC sparing strategies. Interestingly, only 27% of our patients diagnosed between 2013 and 2018 received an immunosuppressant, highlighting the routine strategy of using GC alone at first and introducing GC-sparing treatments in relapsing disease or in patients with poor GC tolerance. The recent ACR guidelines recommend using a combination of GC and tocilizumab rather than GC alone in new-onset GCA [2], and discontinuing GC as soon as possible. This highlights how therapeutic management of GCA is changing and should now include a shorter GC duration and more frequent and earlier use of immunosuppressants, especially at GCA diagnosis when GC doses are high.

Since immunosuppressants were started in our study in patients with relapsing disease or with AEs, who had already reached high GC cumulative doses, we failed to demonstrate a GC-sparing effect and a reduction in GC-related AEs in these patients.

This study took advantage of the patients’ medical files, allowing precise collection of GC therapy information and related AEs. The main limitation of our study regards the AEs not mentioned in the follow-up or hospitalization reports. Moreover, some GC-related AEs are directly treated by general practitioners. Moreover, we analysed and attributed to GC some medical events that might be independent of GC therapy. GCA patients are old and naturally exposed to geriatric complications, independent of GC exposure; since there is no control group, definite conclusions about GC toxicity in the oldest patients cannot be made. Finally, the method we used to calculate the GC cumulative doses may underestimate some results. 

## 5. Conclusions

To date, GC remain mandatory for GCA but their efficiency should not occult their toxicity. Further studies are needed to improve GC management and identify alternative therapeutic strategies that could help drastically reduce GC duration.

## Figures and Tables

**Figure 1 jcm-11-01034-f001:**
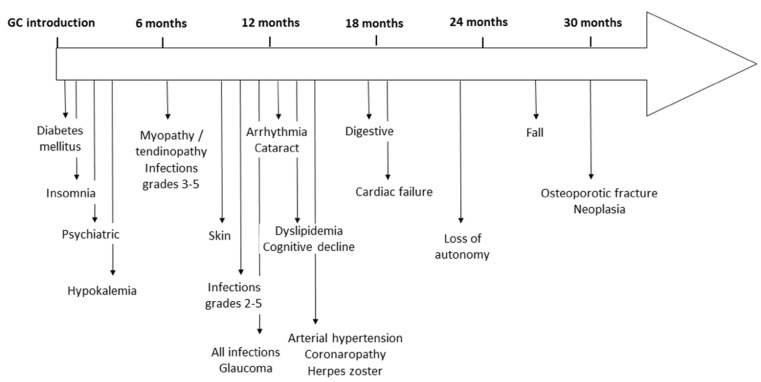
Median time of occurrence of each glucocorticoid-related adverse event once treatment started.

**Table 1 jcm-11-01034-t001:** Characteristics, treatments and outcomes of included GCA patients.

	GCA Patients (*n* = 139)
Female	102 (73)
Age at diagnosis, in years	72 (52–92)
Weight at diagnosis, in kg	62 (45–104)
Body mass index, in kg/m^2^	23.9 (17.2–35.6)
**Past medical history**	
Smoking	32 (23)
Arterial hypertension	65 (47)
Diabetes mellitus	15 (11)
Dyslipidaemia	49 (35)
Coronaropathy	11 (8)
Polymyalgia rheumatica before GCA	14 (10)
**Clinical manifestations at diagnosis**	
Cranial symptoms	112 (81)
Ophthalmologic signs	40 (29)
Polymyalgia rheumatica	51 (37)
**Paraclinical work-up**	
C-reactive protein, in mg/L	75 (3–424)
Erythrocyte sedimentation rate, in millimetre	63 (16–138)
Positive temporal artery biopsy	86 (62)
Large-vessel vasculitis on imaging	40/126 (32)
**Follow-up management**	
Follow-up duration, in months	35.6 (2–111)
Relapses	48 (35)
**Glucocorticoid and immunosuppressant management**	
Initial intravenous boli of solumedrol	25 (18)
Cumulative doses of GC boli, in mg	1250 (75–3750)
Initial oral prednisone dose, in mg/day	50 (10–90)
Duration to reach a dose < 20 mg/d, in months	5 (0.6–89)
Not reached	5 (4)
Duration to reach a dose < 10 mg/d, in months	12 (2–92)
Not reached	8 (6)
Duration to reach a dose < 5 mg/d, in months	20 (5–96)
Not reached	27 (19)
Duration to definitely discontinue GC, in months	22 (8–100)
Not reached	42 (30)
Dose at last follow-up if ongoing GC, in mg/day	5 (1–30)
Total cumulative GC doses at last follow-up in all patients, in milligrams	9184 (1770–24,640)
Total cumulative GC doses at last follow-up in patients who stopped GC, in milligrams	8231 (2836–24,383)
Use of an immunosuppressant	37 (27)

Values are numbers (percentage) or medians [range] GCA: giant cell arteritis; GC: glucocorticoids.

**Table 2 jcm-11-01034-t002:** Glucocorticoid-related adverse events in GCA patients according to whether they received less than or more than 9 g of GC cumulative doses.

	Total (*n* = 139)	Cumulative Dose < 9 g (%) (*n* = 65)	Cumulative Dose ≥ 9 g (%) (*n* = 74)	*p*
**Any AEs**	131 (94)	59 (91)	72 (97)	0.15
**Cardiovascular events**	71 (51)	26 (40)	45 (61)	0.01
Arrhythmia (patients)	21 (15)	6 (9)	15 (20)	0.10
Heart failure	17 (12)	6 (9)	11 (15)	0.44
Arterial hypertension	50 (36)	17 (26)	33 (45)	0.02
Coronaropathy	13 (9)	5 (8)	8 (11)	0.57
Stroke	2 (1)	0	2 (3)	0.50
**Digestive events**	11 (8)	4 (6)	7 (9)	0.54
**Neuropsychiatric events**	66 (47)	33 (51)	33 (45)	0.53
Psychiatric	35 (25)	18 (28)	17 (23)	0.52
Insomnia	47 (34)	21 (32)	26 (35)	0.73
Cognitive impairment	15 (11)	8 (12)	7 (9)	0.60
**Osteomuscular events**	83 (59)	37 (56)	46 (62)	0.46
Osteoporotic fracture	19 (14)	3 (5)	16 (22)	0.004
Myopathy and tendinopathy	39 (28)	19 (29)	20 (27)	0.77
**Geriatric events**	48 (34)	22 (34)	26 (35)	0.87
Falling	39 (28)	18 (28)	21 (28)	0.93
Loss of autonomy	26 (19)	11 (17)	15 (20)	0.61
**Skin fragility**	28 (20)	9 (14)	19 (26)	0.09
**Ophthalmologic events**	33 (24)	12 (18)	21 (28)	0.17
Chronic glaucoma	11 (8)	8 (12)	3 (4)	0.11
Cataract	26 (19)	7 (11)	19 (26)	0.03
**Metabolic events**	87 (63)	36 (55)	51 (69)	0.10
Diabetes mellitus	23 (17)	13 (20)	10 (14)	0.30
Weight gain	71 (51)	27 (42)	44 (59)	0.04
**Hypokalaemia**	23 (17)	9 (14)	14 (19)	0.50
**Infections**	88 (63)	34 (52)	54 (73)	0.01
Herpes zoster	11 (8)	3 (5)	8 (11)	0.22
**Neoplasia**	16 (12)	7 (11)	9 (12)	0.80

Values are numbers (percentage) or medians [range]; GCA: giant cell arteritis; GC: glucocorticoid.

**Table 3 jcm-11-01034-t003:** Glucocorticoid-related AEs in GCA patients according to three groups of GC cumulative doses.

	<8 g (*n* = 50)	8–12 g (*n* = 50)	>12 g (*n* = 39)	*p*
**Cardiovascular events**	22 (44)	28 (56)	21 (54)	0.45
Arrhythmia	6 (12)	7 (14)	8 (21)	0.52
Heart failure	6 (12)	5 (10)	6 (15)	0.74
Arterial hypertension	13 (26)	22 (44)	15 (38)	0.16
Coronaropathy	5 (10)	5 (10)	3 (8)	0.92
**Digestive events (haemorrhage; ulcer; gastric pain)**	3 (6)	3 (6)	5 (13)	0.41
**Neuropsychiatric events**	26 (52)	23 (46)	17 (44)	0.71
Psychiatric	16 (32)	8 (16)	11 (28)	0.16
Insomnia	17 (34)	17 (34)	13 (33)	0.99
Cognitive impairment	6 (12)	6 (12)	3 (8)	0.76
**Osteomuscular events**	28 (56)	32 (64)	23 (59)	0.71
Osteoporotic fracture	1 (2)	9 (18)	9 (23)	0.009
Myopathy and tendinopathy	16 (32)	14 (28)	6 (15)	0.65
**Geriatric events**	19 (38)	15 (30)	14 (36)	0.69
Falling	15 (30)	13 (26)	11 (28)	0.91
Loss of autonomy	11 (22)	7 (14)	8 (21)	0.56
**Skin fragility**	8 (16)	9 (18)	11 (28)	0.32
**Ophthalmologic events**	8 (16)	12 (24)	13 (33)	0.16
Chronic glaucoma	5 (10)	3 (6)	3 (8)	0.76
Cataract	6 (12)	9 (18)	11 (28)	0.15
**Metabolic events**	27 (54)	34 (68)	26 (67)	0.29
Diabetes mellitus	9 (18)	7 (14)	7 (18)	0.83
Weight gain	20 (40)	28 (56)	23 (59)	0.14
**Hypokalaemia**	6 (12)	11 (22)	6 (15)	0.39
**Infections**	27 (54)	31 (62)	30 (77)	0.08
**Herpes zoster**	3 (6)	3 (6)	5 (13)	0.40
**Neoplasia**	6 (12)	6 (12)	4 (10)	0.96

Values are numbers (percentage) or medians [range]. GCA: giant cell arteritis; GC: glucocorticoid.

**Table 4 jcm-11-01034-t004:** Glucocorticoid-related AEs in GCA patients according to three groups of GC durations.

	<20 Months (*n* = 47)	20–40 Months (*n* = 50)	>40 Months (*n* = 42)	*p*
**Cardiovascular events**	18 (38)	27 (54)	26 (62)	0.07
Arrhythmia (patients)	6 (13)	5 (10)	10 (24)	0.16
Heart failure	6 (13)	6 (12)	5 (12)	0.99
Arterial hypertension	12 (26)	19 (38)	19 (45)	0.14
Coronaropathy	3 (6)	7 (14)	3 (7)	0.37
**Digestive events (haemorrhage; ulcer; gastric pain)**	1 (2)	4 (8)	6 (14)	0.11
**Neuropsychiatric events**	21 (45)	24 (48)	21 (50)	0.88
Psychiatric	12 (26)	12 (24)	11 (26)	0.97
Insomnia	12 (26)	19 (38)	16 (38)	0.34
Cognitive impairment	5 (11)	6 (12)	4 (10)	0.93
**Osteomuscular events**	25 (53)	31 (62)	27 (64)	0.52
Osteoporotic fracture	5 (11)	4 (8)	10 (24)	0.07
Myopathy and tendinopathy	11 (23)	18 (36)	10 (24)	0.31
**Geriatric events**	17 (36)	17 (34)	14 (33)	0.96
Falling	15 (32)	12 (24)	12 (29)	0.68
Loss of autonomy	7 (15)	10 (20)	9 (21)	0.70
**Skin fragility**	8 (17)	9 (18)	11 (26)	0.50
**Ophthalmologic events**	9 (19)	7 (14)	17 (40)	0.008
Chronic glaucoma	5 (11)	2 (4)	4 (10)	0.43
Cataract	7 (15)	5 (10)	14 (33)	0.01
**Metabolic events**	24 (51)	34 (68)	29 (69)	0.13
Diabetes mellitus	7 (15)	11 (22)	5 (12)	0.40
Weight gain	17 (36)	27 (54)	27 (64)	0.03
**Hypokalaemia**	5 (11)	11 (22)	7 (17)	0.32
**Infections**	24 (51)	32 (64)	32 (76)	0.048
Herpes zoster	3 (6)	0	8 (19)	0.003
**Neoplasia**	2 (4)	7 (12)	7 (17)	0.15

Values are numbers (percentage) or medians [range]; GCA: giant cell arteritis; GC: glucocorticoid.

**Table 5 jcm-11-01034-t005:** Adverse events in GCA patients according to whether they received an immunosuppressant.

	Immunosuppressant (*n* = 37)	No Immunosuppressant (*n* = 102)	*p*
**Cardiovascular events**	23 (62)	48 (47)	0.13
Arrhythmia (patients)	11 (30)	10 (10)	0.004
Heart failure	4 (11)	13 (13)	1.00
Arterial hypertension	16 (43)	34 (33)	0.28
Coronaropathy	5 (14)	8 (8)	0.33
Stroke	0	2 (2)	1.00
**Digestive events (haemorrhage; ulcer; gastric pain)**	4 (11)	7 (7)	0.48
**Neuropsychiatric events**	20 (54)	46 (45)	0.35
Psychiatric	12 (32)	23 (23)	0.24
Insomnia	13 (35)	34 (33)	0.84
Cognitive impairment	4 (11)	11 (11)	1.00
**Osteomuscular events**	23 (62)	60 (58)	0.68
Osteoporotic fracture	7 (19)	12 (12)	0.28
Myopathy and tendinopathy	12 (32)	27 (26)	0.49
**Geriatric events**	13 (35)	35 (34)	0.93
Falling	12 (32)	27 (26)	0.49
Loss of autonomy	5 (14)	21 (21)	0.34
**Skin fragility**	8 (22)	20 (20)	0.79
**Ophthalmologic events**	9 (24)	24 (24)	0.92
Chronic glaucoma	2 (5)	9 (9)	0.73
Cataract	8 (22)	18 (18)	0.63
**Metabolic events**	23 (62)	64 (63)	0.95
Diabetes mellitus	3 (8)	20 (20)	0.13
Weight gain	19 (51)	52 (51)	0.99
**Hypokalaemia**	6 (16)	17 (17)	0.99
**Infections**	27 (73)	61 (60)	0.15
Herpes zoster	4 (11)	7 (7)	0.48
**Neoplasia**	4 (11)	12 (12)	1.00

Values are numbers (percentage) or medians [range]; GCA: giant cell arteritis; GC: glucocorticoid.

## Data Availability

The data included in this article cannot be shared publicly due to the requirement to maintain the privacy of the individuals who participated in the study. The data will be shared upon reasonable request to the corresponding author.

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
