# Peer review of "Impact of Glucocorticoid Cumulative Doses in a Real-Life Cohort of Patients Affected by Giant Cell Arteritis"

_jcm, 2022, doi:10.3390/jcm11041034_

Round 1

Reviewer 1 Report

This is a very interesting manuscript on real-life data. Some points should be considered to provide additional informations from the data:

Major comments:

Ad Methods: How many patients had a short follow-up of less than 1 week? These patients should not be included into the study. 

Ad results: I would like to see an additional figure with Kaplan Meier curves comparing patients with and without immunosuppression, to show the time points of first diagnosis of cardiovascular events, neuropsychiatric events and osteoporotic fractures (out of the patients included into the study at this time point). This information could provide insights into the need of searching for these side effects in the management of these patients. 

ad Discussion:

  • The use of MTX and tocilizumab in this cohort is limited to 26% of the patients. This aspect is important to be discussed in view of the 2018 and the earlier EULAR recommendations for the management of large vessel vasculitis.
  • The Kaplan Meier curves should be discussed with their relevance for the management of these patients.

Ad Conclusions: Bec cause of the frequent side effects shown in this manuscript, the conclusion should not support glucocorticoids (GCs) as the first-line treatment for GCA. This manuscript rather supports the need for alternative treatments instead of or at least in addition to GCs. 

Minor comment:

Ad Table 1: The legend suggests characteristics at baseline, but includes data on relapses and CS management. I propose to show all data on relapses and CS in a separate table.

Author Response

Response to Reviewer 1

- Methods: How many patients had a short follow-up of less than 1 week? These patients should not be included into the study. 

Thank you for this pertinent comment.

One patient indeed had a short follow-up of less than 1 week (diagnosed in our department and followed-up elsewhere).

We excluded this patient.

Data in the Tables were thus modified with a cohort of 139 patients.

Our results are unchanged after excluding this patient.

- Results: I would like to see an additional figure with Kaplan Meier curves comparing patients with and without immunosuppression, to show the time points of first diagnosis of cardiovascular events, neuropsychiatric events and osteoporotic fractures (out of the patients included into the study at this time point). This information could provide insights into the need of searching for these side effects in the management of these patients. 

Thank you for this idea.

We created the 3 following figures showing Kaplan Meier curves analyzing occurrence of cardiovascular events, neuropsychiatric events and osteoporotic fractures in patients with or without an immunosuppressant.

Unfortunately, we did not demonstrate any difference in both groups and finally decided not to include these figures in the manuscript.

We failed to demonstrate a GC sparing effect and a reduction of GC-related AEs in patients who received an immunosuppressant because French and European recommendations currently advise us to introduce a GC-sparing treatment in patients with relapsing disease or poor GC tolerance. These patients thus received an immunosuppressant at a later time in the disease course when cumulative GC doses were already high. This explains why we tend to observe more GC-related AEs in patients with an immunosuppressant (see Table 5).

In the result section, in paragraph 3.5 Impact of immunosuppressant use on adverse event occurrence, we wrote: " In addition, we did not observe a lower rate of GC-related AEs in patients who received an immunosuppressant."

In the discussion, we wrote: "Since immunosuppressants were started in our study in patients with relapsing disease or with AEs, who had already reached high GC cumulative doses, we failed to demonstrate a GC sparing effect and a reduction of GC-related AEs in these patients."

- Discussion:

The use of MTX and tocilizumab in this cohort is limited to 26% of the patients. This aspect is important to be discussed in view of the 2018 and the earlier EULAR recommendations for the management of large vessel vasculitis.

We agree with this interesting point. European and French recommendations advise to use immunosuppressants in patients with relapsing disease or in patients in whom GC tolerance is problematic.

Interestingly, the patients from this study were diagnosed with GCA between 2013 and 2018. At the time, immunosuppressants were often added in patients with GC-dependent disease or when adverse events had occurred. Since the publication of the tocilizumab trial in 2017, the therapeutic approach has changed and now includes an earlier consideration of GC-sparing strategies. In our center, since 2018, we now prescribe immunosuppressants more frequently to try to reduce the duration of exposure to GC. Another reflection of the changes that have occurred in the last few years is in the recent ACR recommendations (Maz M et al, Arthritis Rheumatol, 2021). The optimal GC duration is not indicated assuming clinicians will try to discontinue them as early as possible. In addition, first-line treatment should now combine GC and tocilizumab in new-onset GCA, in order to reduce the likelihood of GC adverse events.

Our real-life study is a reminder that the old (but current) European and French recommendations to treat GCA only with GC should probably be reconsidered.

We added in the discussion, before the limitations’ section:

" Our patients were treated in accordance with French and European guidelines, recommending the use of GC for at least 18 months [3,20]. With a high rate of GC-related AEs in our patients, this study underlines the urgent necessity of GC sparing strategies. Interestingly, only 27% of our patients diagnosed between 2013 and 2018 received an immunosuppressant, highlighting the routine strategy of using GC alone at first and introducing GC-sparing treatments in relapsing disease or in patients with poor GC tolerance. The recent ACR guidelines recommend using the combination of GC and tocilizumab rather than GC alone in new-onset GCA [2], and discontinuing GC as soon as possible. This highlights how therapeutic management of GCA is changing and should now include a shorter GC duration and more frequent and earlier use of immunosuppressants, especially at GCA diagnosis when GC doses are high."  

The Kaplan Meier curves should be discussed with their relevance for the management of these patients.

As discussed earlier, our patients already had important cumulative doses of GC at the time of initiation of the immunosuppressant. We thus failed to demonstrate a GC sparing effect and a reduction of GC-related AEs in these patients.

In the discussion, we wrote: "Since immunosuppressants were started in our study during follow-up in patients with relapsing disease or with AEs, who had already reached high GC cumulative doses, we failed to demonstrate a GC sparing effect and a reduction of GC-related AEs in these patients."

Conclusions: Because of the frequent side effects shown in this manuscript, the conclusion should not support glucocorticoids (GCs) as the first-line treatment for GCA. This manuscript rather supports the need for alternative treatments instead of or at least in addition to GCs.

We agree. We modified the sentence as follows: "To date, GC remain mandatory for GCA but their efficiency should not occult their toxicity. Further studies are needed to improve GC management and identify alternative therapeutic strategies that could help drastically reduce GC duration".

Table 1: The legend suggests characteristics at baseline, but includes data on relapses and CS management. I propose to show all data on relapses and CS in a separate table.

Thank you for this careful analysis. Since the article already contains 5 Tables and 1 figure, we renamed Table 1 : "Characteristics, treatments and outcomes of included GCA patients"

Reviewer 2 Report

Dear Authors,

I read with interest your manuscript focused on a hot topic in the field large vessel vasculitides, although it has been already addressed in several previous study.

The major strengthens of this the study is the completeness and precision of the performed analysis, that may add significant value to this manuscript compared with previous studies.

I think it is well written and easy to read. Methods are adequately described, and the results are clearly presented.

I have few suggestions/comments that hopefully may contribute to further improve the quality of this paper:

  • Include a stratified analysis, where AEs are distinguished in "surely" and "possibly" related to GCs.  This data may allow to better interpretate the very high prevalence of "any GC-AE" of 95%.
  • In the discussion authors may comment whether they think the current recommendation on the management of GC are adequate or a revision is needed
  • In the discussion might be better explain which are the practical consequence from the result especially in regard to the need of GC-sparing agents

Author Response

Include a stratified analysis, where AEs are distinguished in "surely" and "possibly" related to GCs.  This data may allow to better interpretate the very high prevalence of "any GC-AE" of 95%.

Thank you for this idea.

We agree that this distinction might be useful to focus mainly on "surely" GC-related AEs.

When analyzing our database, we experienced some difficulties trying to classify some AEs (fall, amyotrophy, cataract, hypertension...). In our patients, many other factors might contribute to the AE, such as age, other medications, geriatric situations, etc.

We thus failed to produce a relevant distinction and the classification we did was probably confusing and not reproductible.

Here are two examples:

- Occurrence of a cataract in an 87 year-old women receiving 8 grams of prednisone is “possibly” or “surely” linked to GC. However, at this age, cataract is frequent, even in patients without GC.

- Occurrence of a fall or infection in an elderly person can “surely” or “possibly” be linked to GC.

Hence, to include this fitting remark and to modulate the attribution of all AEs to GCs alone, we added in the limitations section in the Discussion: "Moreover, we analyzed and attributed to GC some medical events that might be independent of GC therapy. GCA patients are old and naturally exposed to geriatric complications, independent of GC exposure; since there is no control group, definite conclusions about GC toxicity in the oldest patients cannot be made."

In the discussion authors may comment whether they think the current recommendation on the management of GC are adequate or a revision is needed

We believe our study highlights the fact that GC tolerance remains an important issue in routine clinical practice. Our patients were treated in accordance with French and European guidelines, which recommend treating GCA with GC for at least 18 months. These guidelines are currently very different than those of the ACR (Maz M et al, Arthritis Rheumatol, 2021): the optimal GC duration is not indicated assuming clinicians will try to discontinue them as early as possible. In addition, first-line treatment should now combine GC and tocilizumab in new-onset GCA, in order to reduce the likelihood of GC adverse events.

In the discussion, before the limitation section, we wrote:

"Our patients were treated in accordance with French and European guidelines, recommending the use of GC for at least 18 months [3,20]. With a high rate of GC-related AEs in our patients, this study underlines the urgent necessity of GC sparing strategies. Interestingly, only 27% of our patients diagnosed between 2013 and 2018 received an immunosuppressant, highlighting the routine strategy of using GC alone at first and introducing GC-sparing treatments in relapsing disease or in patients with poor GC tolerance. The recent ACR guidelines recommend using the combination of GC and tocilizumab rather than GC alone in new-onset GCA [2], and discontinuing GC as soon as possible. This highlights how therapeutic management of GCA is changing and should now include a shorter GC duration and more frequent and earlier use of immunosuppressants, especially at GCA diagnosis when GC doses are high.”  

In the discussion might be better explain which are the practical consequence from the result especially in regard to the need of GC-sparing agents

As discussed in the previous query, our study highlights how the recent ACR recommendations are relevant as they place immunosuppressants at GCA onset with a shortened GC therapy.

We have added this to our discussion as mentioned above.
